# Vitiligo-like Lesions and COVID-19: Case Report and Review of Vaccination- and Infection-Associated Vitiligo

**DOI:** 10.3390/vaccines10101647

**Published:** 2022-09-30

**Authors:** Laura Macca, Lucia Peterle, Manuela Ceccarelli, Ylenia Ingrasciotta, Giuseppe Nunnari, Claudio Guarneri

**Affiliations:** 1Department of Clinical and Experimental Medicine, Section of Dermatology, University of Messina, Messina, Italy C/O A.O.U.P. “Gaetano Martino”, Via Consolare Valeria, 98125 Messina, Italy; 2Department of Clinical and Experimental Medicine, Unit of Infectious Diseases, University of Catania, Catania, Italy C/O ARNAS “Garibaldi”, “Nesima” Hospital, Via Palermo 636, 95122 Catania, Italy; 3Department of Biomedical and Dental Sciences and Morphofunctional Imaging, Section of Pharmacology, University of Messina, Messina, Italy C/O A.O.U.P. “Gaetano Martino”, Via Consolare Valeria, 98125 Messina, Italy; 4Department of Clinical and Experimental Medicine, Unit of Infectious Diseases, University of Messina, Messina, Italy C/O A.O.U.P. “Gaetano Martino”, Via Consolare Valeria, 98125 Messina, Italy; 5Department of Biomedical and Dental Sciences and Morphofunctional Imaging, Section of Dermatology, University of Messina, Messina, Italy C/O A.O.U.P. “Gaetano Martino”, Via Consolare Valeria, 98125 Messina, Italy

**Keywords:** autoimmunity, autoimmune diseases, vitiligo, vaccine, COVID-19

## Abstract

Several cutaneous manifestations in patients undergoing COVID-19 vaccination have been described in the literature. Herein, we presented a case of new-onset vitiligo that occurred after the second dose of the Comirnaty vaccine. An updated literature search revealed the occurrence of a total of 16 cases, including new-onset or worsening of preexisting vitiligo. Given the autoimmune pathogenesis of the disease, we reviewed and discussed the potential role of the vaccine prophylaxis as a trigger for the development of such hypopigmented skin lesions.

## 1. Introduction

Since the beginning of the coronavirus (COVID-19) vaccination campaign, several skin adverse reactions after the dose administrations have been reported in the medical literature [1,2,3]. They mainly consist of delayed inflammatory reactions at the injection site, urticaria, chilblain-like lesions, and pityriasis rosea-like eruptions [2,4]. However, cutaneous and extracutaneous autoimmune diseases (ADs) have been documented [1]. Herein, we presented a case report of a patient who developed new-onset vitiligo after receiving the second dose of the Comirnaty vaccine.

Although a clear etiopathogenetic link between SARS-CoV-2 infection with related prophylaxis and this typical cutaneous autoimmune disease has not been definitely demonstrated, their association emerged during the pandemic and postpandemic era and seems to be not accidental. With the purpose of reviewing the pertinent literature, we checked the PubMed (https://ncbi.nlm.nih.gov/PubMed, accessed on 15 August 2022), Scopus, and Web of Science databases using the string “Vitiligo” [All Fields] AND “COVID-19” [All Fields] without time limits. According to the search results and their critical analysis, we discussed possible hypotheses that may underlie such unexpected events.

## 2. Case Presentation

A 35-year-old Caucasian woman presented to the dermatology clinic complaining of sudden depigmentation of the hands. The patient reported the occurrence of discoloration after receiving the second dose of the Comirnaty vaccine. Physical examination revealed multiple depigmented patches on bilateral dorsal hands (Figure 1A). The patches were consistent with vitiligo, and evaluation under Wood’s light revealed the characteristic white fluorescence (Figure 1B). No other body areas were involved. Blood test including thyrotropin was within the normal range. Antithyroid peroxidase and antithyroglobulin antibodies were negative. Familial and personal medical history was unremarkable for other cutaneous or extracutaneous autoimmune diseases. Moreover, no history of systemic or topical drugs reported to be linked with secondary vitiligo was reported. COVID-19 vaccination was considered as the potential culprit for the skin changes given the lack of other potential causative factors and the temporal link between vaccination and the lesions’ outbreak. She was treated with topical khellin twice daily. The outcome at 3 months was not available because the patient was lost to follow-up. A written consent to treatment and image recording for academic purposes was obtained.

## 3. Review of Literature

To the best of our knowledge, only 16 additional cases linking vitiligo to COVID-19 have been reported in the literature and are summarized in Table 1, including the ones occurring as a new onset and recrudescence after vaccination and manifestations of the disease in the course of COVID-19 infection [1,5,6,7,8,9,10,11,12,13,14,15,16,17,18,19]. Our search revealed 7 male patients affected versus 11 female patients whose age at presentation ranged from 13 to 86 years (mean age 50.625 ± 17.421). Out of 16, 2 (one man and one woman) had preexisting vitiligo, while the rest of the patients newly presented hypopigmented lesions after vaccination. The onset of the condition generally occurred a few days (maximum 1 week) after the dose administration: in seven cases, lesions abrupted after the first dose; in four cases after the second dose; and in three cases, they were repetitive after the first and second doses. In two cases, vitiligo appeared after the COVID-19 disease. Out of fourteen vaccinated patients, nine presented the skin condition after Comirnaty administration, whereas three after Moderna and only two after AstraZeneca. The three cases characterized by the recurrence of lesions at the second dose administration occurred with each of the three different types of vaccination. With regard to the clinical manifestations, vitiligo presented with the classical topography except for one case, in which hypopigmentation occurred at the site of injection as the isomorphic response. According to familial anamnesis, only two patients reported a history of the skin disease. Curiously, these patients were not positive for preexisting lesions. Manifestations remained generally stable, limited to the observation time, with poor or no response to the dermatological treatments and with only four cases slightly improving with calcineurin inhibitors and/or UV phototherapy plus topical or oral steroids. In one case, marked by preexisting lesions, worsening was noted. Autoimmunity- and autoinflammatory-based comorbidities, including ulcerative colitis, psoriasis, Hashimoto’ thyroiditis, type II diabetes mellitus, and ankylosing spondylitis, were present in four patients, whereas hypothyroidism and slight positivity of antinuclear antibodies (ANAs) were recorded in additional two patients. The unique case of Koebnerization at the inoculation site occurred in a patient having psoriasis, Hashimoto’s thyroiditis, and type II diabetes.

## 4. Discussion

Vitiligo is a common autoimmune skin disease involving approximately 0.5–2% of the world population [20], ranging from 0.38% in the Danish peninsula and 0.49% in the USA to 1.9% in China and 2.28% in a remote area of Romania, with no differences between adults and children [21].

The onset of vitiligo often occurs in younger individuals and progresses with time, resulting in a disfiguring disease [22]. However, the clinical course remains generally unpredictable in both segmental and nonsegmental form [23]. The characteristic milky-white macules derive from the selective destruction of melanocytes in the skin or hair or both [22]. Recent insights into the pathogenesis of vitiligo offer a better understanding of the course of this autoimmune disease. On the background of genetic susceptibility, the presence of an intrinsic anomaly of melanocytes makes them more sensitive to oxidative damage, leading to a greater expression of proinflammatory proteins such as Heat Shock Protein 70 kilodaltons (HSP70), which plays a pivotal role in the promotion of specific immune responses assisting the melanocyte-derived peptides’ uptake, processing, and presentation to major histocompatibility complex (MHC) [24,25]. Moreover, the lower expression of epithelial adhesion molecules, such as discoidin domain receptor 1 (DDR1) and E-cadherin, enhances melanocytes’ damage and antigen exposure resulting in the promotion of autoimmunity [26]. Stressed vitiligo melanocytes release exosomes and inflammatory cytokines that lead to innate immune response induction and, subsequently, to adaptive immune response through the activation of autoreactive cytotoxic CD8+ T cells [27]. The latter produce interferon-γ (IFN-γ) leading to disease progression through IFN-γ-induced chemokine secretion from surrounding keratinocytes, which results in the recruitment of more T cells to the skin through a positive feedback loop [27]. The described activation of the IFN pathway perpetuates the direct action of CD8+ cells against melanocytes, facilitated by regulatory T-cell dysfunction [27]. Recent scientific advances have also led to a deeper understanding of the complex role played by a specific subtype of T cells: T-resident memory cells [28]. Indeed, CD8 tissue-resident memory T cells are responsible for long-term maintenance and potential relapse of vitiligo in patients through cytokine-mediated T-cell recruitment from the bloodstream [27]. Hence, Koebner’s phenomenon may be explained by the higher frailty of altered melanocytes, which results in the release of inflammatory mediators, production of reactive oxygen species (ROS), and melanocyte death in response to any skin trauma [26].

On the other hand, with regard to immune response activation and SARS-CoV-2 pathogenesis, a central role is played by angiotensin converting enzyme (ACE) 2, mainly expressed in bronchial transient secretory cells and in type 2 alveolar cells [29]. In fact, it has been identified as the receptor molecule for the cellular entry of the virus, so when SARS-CoV-2 enters the body, the spike protein binds to the host cell through ACE2 allowing the virus to fuse with the cell membrane and to release the viral RNA into the cytoplasm [29]. The viral invasion of host cells promotes type-I interferon (IFN- α/β) and proinflammatory cytokine synthesis and secretion [30]. Spreading of the virus inside the body is contained by the action of antigen-presenting macrophage and natural killer (NK) cells [30]. Interestingly, the inhibition of IFN production by the N protein of SARS-CoV-2 represents a crucial stage for viral survival, whereas Th1-type immune response has a key role in virus clearance promotion [29]. Indeed, helper T cells promote the nuclear factor kappa-light-chain-enhancer of activated B-cell (NF-kB) signaling pathway activation that leads to proinflammatory cytokine production [29]. Helper T cells also promote viral-specific antibodies secretion through the activation of T-dependent B lymphocytes [31]. Cytotoxic T cells directly kill virus-infected cells [29]. After the invasion of the respiratory mucosa, the virus particles can infect other cells that possess the specific receptor, triggering a series of immune responses and producing a “cytokine storm” driven by IL-2, IL-6, IL-7, IL-10, C-reactive protein (CRP), granulocyte colony-stimulating factor (G-CSF), macrophage colony-stimulating factor (MCSF), interferon-gamma inducible protein (IP) 10, monocyte chemoattractant protein (MCP) 1, macrophage inflammatory protein (MIP) 1α, IFN-γ, and tumor necrosis factor (TNF) α [30]. The “cytokine storm”, with the aim of reducing the viral spread, initiates inflammatory-induced lung injury with life-threatening complications such as multiorgan failure (MOF), acute respiratory distress syndrome (ARDS), septic shock, hemorrhage and thrombosis, acute heart/liver/kidney injury, and secondary bacterial infections [29]. As of 7 September 2022, there have been 603,711,760 COVID-19 confirmed cases, including 6,484,136 deaths globally reported to the World Health Organization (WHO). As of 4 September 2022, a total of 12,540,061,501 vaccine doses have been administered [31].

Since the COVID-19 outbreak has changed our health programs in the last months, vaccination has become a daily practice. However, the temporal relationship between the vaccine and development of some skin conditions makes vitiligo development after prophylaxis more than just a coincidence [1,32]. Based on genetic susceptibility, some patients presented with different reactions linked with vaccine administration. Possible pathogenesis remains unclear, although the immune mechanisms underlying both the cutaneous conditions and action of the drug have been called into question, including antigen presentation, cytokine production, epitope spreading, and polyclonal activation of B cells involved in both the anti-infectious immune response as well as in autoreactivity [1,10,33]. In particular, with regard to vitiligo, melanocytes may have represented an unintentional target for antibodies and immune cell responses [10]. Given that the development of vitiligo involves the destruction of melanocytes by autoreactive CD8+ T cells, and successful vaccination also involves an extensive CD8+ T-cell response, this hypothesis may be plausible [10]. Another triggering mechanism may be the production of type I interferons (IFN-I) by plasmacytoid dendritic cells (pDCs) caused by vaccination [34,35]. Indeed, IFN-I and pDCs play a crucial role both in the defense against SARS-CoV-2 and the activation of the immune response in vitiligo [34,35]. Finally, the IFN-I-mediated immune response induced by vaccination may be linked to the concurrent development of autoimmune disorders such as vitiligo.

Similarly, dealing with cutaneous autoimmune diseases, psoriasis flares, both de novo onset and recrudescence, has been described [36,37,38]. The possible role of viral proteins and vaccine adjuvants as triggers for immune dysregulation has been assumed, leading to the inflammatory cascade through IL-6 production and recruitment of Th17 cells [32,39]. No differences in the type of causing vaccine or correlations with the type of psoriasis were recorded, whereas variable response to treatment and clinical outcomes resulted [36,37,38].

In particular, a generalized pustular variant of psoriasis is driven by IFN-I itself, thus supporting the above-mentioned possible understanding of the association [38,40].

Anyway, with adverse cutaneous reactions, the collection of data from large case series is fundamental, allowing for further deep studies and more reasonable explanations.

Our contribution through reviewing the medical literature with regard to such uncommon events represents a useful guidance for clinicians to detect and correctly manage these conditions, especially in times of unpredictable medical enemies.

## 5. Conclusions

As the reporting of autoimmune reactions following COVID-19 vaccination continues to expand, it is reasonable to include it in the list of possible triggers of de novo vitiligo lesions. A link between inflammatory cells involved in both the pathogenesis of vitiligo and mechanism by which the COVID-19 vaccination stimulates the immune system may be assumed, and physicians should be aware of such skin reactions to vaccinations, especially when dealing with genetically susceptible patients.

## Figures and Tables

**Figure 1 vaccines-10-01647-f001:**
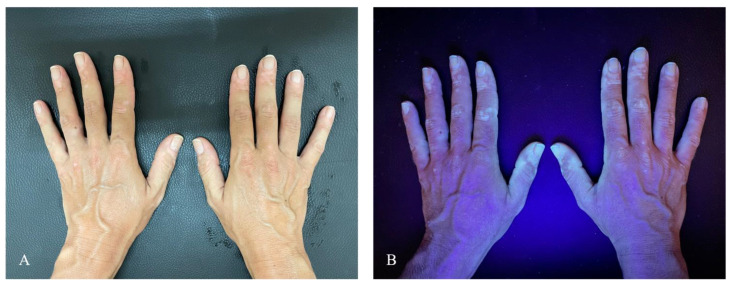
(**A**) Multiple depigmented patches on bilateral dorsal hands. (**B**) Evidence of hypopigmented areas under Wood’s light.

**Table 1 vaccines-10-01647-t001:** Synoptical review of cases of vitiligo after COVID-19 and COVID-19 vaccination reported in medical literature.

Authors, Reference Number, and Year	Type of Study	Sex/Age (Year)	Country of Origin	Clinical Course	Onset after Vaccination	Localization	Vitiligo Familial History	Comorbidities	Treatment	Outcome
**Aktas H** [5], 2021	Case report	M/58	Turkey	New onset after Pfizer- BioNTech first dose	1 week	Face	No	Ulcerative colitis	Topical tacrolimus	1 month: stable
**Kaminetsky J** [6], 2021	Case report	W/61	USA	New onset after Moderna mRNA-1273 first and second dose	Several days after first dose, progressing after second dose	Anterior neck after first dose, spreading to face, neck, chest, abdomen after second dose	No	None	Topical calcineurin inhibitor, phototherapy	Not reported
**Ciccarese G** [1], 2021	Case report	W/33	Italy	New onset after Pfizer- BioNTech first dose	1 week	Trunk, neck, back	Yes (father)	None, ANA + (1:160) nucleolar pattern	Antioxidants systemic, heliotherapy	1 month: stable
**Herzum A** [7], 2021	Case report	W/45	Italy	New onset after COVID-19	2 weeks	Limbs, face, trunk	No	None	Phototherapy	1 month: stable
**Lopez Riquelme I** [8], 2022	Case report	W/60	Spain	New onset after AstraZeneca first dose	3 days	Face and arms	Not reported	None	Topical tacrolimus	Not reported
**Flores-Terry M** [9], 2022	Case report	W/39	Spain	New onset after Pfizer- BioNTech second dose	1 week	Face	No	None	Not reported	Not reported
**Militello M** [10], 2022	Case report	W/67	USA	New onset after Moderna mRNA-1273 first dose	2 weeks	Dorsal hands	Not reported	None	Topical steroids	Not reported
**Nicolaidou E** [11], 2022	Case report	M/69	Greece	New onset after Pfizer- BioNTech first and second dose	Few days after first dose, progressing after second dose	Face, abdomen, back, upper and lower limbs	No	None	Phototherapy	2 months: partial improvement
**Singh R** [12], 2022	Case report	W/43	USA	New onset after Moderna mRNA-1273 first dose	3 weeks	Left upper arm (at the site of injection), face, scalp, chest	Not reported	Hashimoto’s thyroiditis, type II diabetes, psoriasis	Topical steroids and topical tacrolimus	Not reported
**Schmidt A** [13], 2022	Case report	W/52	USA	New onset after COVID-19	4 weeks	Neck, face, trunk	No	Hypothyroidism	Topical tacrolimus, topical and oral steroids	7 months: partial improvement
**Bukhari A** [14], 2022	Case report	W/13	Saudi Arabia	New onset after Pfizer- BioNTech first dose	2 weeks	Upper and lower limbs	Yes (father and paternal uncle)	No	Topical calcineurin inhibitor, topical steroid, and localized phototherapy	3 months: partial improvement
**Uğurer E** [15], 2022	Case report	M/47	Turkey	New onset after Pfizer- BioNTech first dose	1 week	Bilateral axilla and forearm flexor surfaces	Not reported	Ankylosing spondylitis	Topical pimecrolimus	1 month: partial improvement
**Koç Yıldırım** S [16], 2022	Case report	M/49	Turkey	New onset after Pfizer- BioNTech second dose	2 weeks	Face	No	No	Topical tacrolimus	Not reported
**Gamonal S** [17], 2022	Case report	M/86	Brazil	New onset of vitiligo and lichen planus after AstraZeneca first and second dose	1 week after first dose, progressing after second dose	Upper and lower limbs, trunk, and buttocks	Not reported	Not reported	Topical steroid	Not reported
**Caroppo F** [18], 2022	Case report	M/66	Italy	Preexisting vitiligo worsened after Pfizer-BioNTech second dose	2 weeks	Upper and lower limbs, face, trunk, genital area	No	Hashimoto’s thyroiditis	Oral and topical steroids, topical tacrolimus	Not reported
**Okan G** [19], 2021	Case report	M/22	Turkey	Preexisting vitiligo worsened after Pfizer-BioNTech second dose	2 weeks	Face	No	None	Topical tacrolimus	2 months: no response

Legend: W: woman, M: man.

## Data Availability

Not applicable.

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
