# Peer review of "Vitiligo-like Lesions and COVID-19: Case Report and Review of Vaccination- and Infection-Associated Vitiligo"

_vaccines, 2022, doi:10.3390/vaccines10101647_

Round 1

Reviewer 1 Report

The authors have reported vitiligo-like lesions post second dose of COVID-19 vaccination. However, several such studies have already been reported as mentioned in Table-1. What was the novelty of this case report? The authors should have assayed for IL-6, IFN-I levels and /or Th17 cells to ascertain inflammatory cascade activation post COVID-19 vaccination.

Abstract:

1)      Lines 22-23 - The authors mention ‘we present a case of new-onset vitiligo that occurred after 22 the second dose of Comirnaty vaccine’. However, cases of vitiligo post-second dose of Covid-19 vaccination has also been reported as mentioned below (references 18 and 12 in the manuscript) and a few other cases as reported by authors in Table 1.

i)        Caroppo, F., Deotto, M.L., Tartaglia, J. and Fortina, A.B., 2022. Vitiligo worsened following the second dose of mRNA SARSCoV2 vaccine. Dermatologic Therapy35(6).

ii)       Singh, R., Cohen, J.L., Astudillo, M., Harris, J.E. and Freeman, E.E., 2022. Vitiligo of the arm after COVID-19 vaccination. JAAD Case Reports.

Introduction:

2)      Line 30 – Abbreviate ‘COVID-19’ properly.

3)      Lines 31-32 – ‘…dose administrations have been reported in the medical literature’. What do ‘dose administrations’ and ‘medical literature’ imply?

4)      Lines 33-34 – ‘However, cutaneous, and extracutaneous autoimmune diseases (ADs) have rarely been documented’. Again, this statement is incorrect as mentioned in comment (1) above and by authors in Table 1.

Author Response

Dear Reviewer

first of all, we would like to thank you for the valuable comments, aimed to improve the quality of our manuscript. We tried to fit the questions arisen during the review process and we hope that the paper could be considered of interest for the readers of the journal in its current presentation. Please find enclosed our point-to-point reply to the criticisms. We apologize in advance for some misunderstandings in considering some of your comments (e.g. point #1 and #3).

  • Our case presentation represents an additional report on a really uncommon adverse cutaneous reaction after COVID-19 vaccination and the opportunity of an updated review of pertinent literature with discussion on possible pathogenic interpretation and management. Although cases similar to ours have been reported, in our opinion the topic is of interest and a collection of such comprehensive data is still lacking. With regard to the assessment of the case, we hypothesized the causality of the phenomenon on a main clinical basis, similarly to several other reports, in absence of additional assays. Moreover, the patient, overwhelmed by previous hospital staying, refused other laboratory exams.
  • With regard to Abstract, we are not clear with referee’s criticism #1. We stated the description of an additional (not unique) case of new-onset vitiligo after COVID vaccination and included the revision of the two cases mentioned by the referee and other 14 cases found in literature. Furthermore, the report by Caroppo et al. describe a case of pre-existing vitiligo worsening after Comirnaty vaccination.
  • Regarding Introduction: #2 We used COVID-19 just as an abbreviation, according with the whole body of the manuscript and current similar literature. Could you please suggest a possible more proper definition? #3 We are not clear with this criticism, could you please explain how we could improve the sentence? #4 “rarely” has been deleted from the sentence you mentioned.

Reviewer 2 Report

Dear authors,

Thank you very much for your case report and the comprehensive review! It was very informative to read.

Macca et al describe a vitiligo-like lesion that appeared after SARS-CoV-2 vaccination; they furthermore report on results of a literature research on vitiligo associated with SARS-CoV-2 infection or vaccination.

The case report is interesting, however the strength of the paper is the literature review that is comprehensive and complete.

To improve this manuscript, I suggest to make clear in the title, that not only vaccination-associated vitiligo, but also infection-associated are reported in the review part.

Furthermore, the authors should refer to the different prevalence of vitiligo in different regions and countries and complete their literature review with the countries of origin of the individual case reports.

Sear terms used for literature search would be helpful to be included in materials and methods.

Please clarify in both the title and around lines 62/63 that you are reporting a vaccination-associated case of vitiligo, but the literature review includes vaccination-associated and COVID-19-infection-associated cases of vitiligo.

Best wishes!

Author Response

Dear Reviewer

first of all, we would like to thank you for the valuable comments, aimed to improve the quality of our manuscript. We tried to fit the questions arisen during the review process and we hope that the paper could be considered of interest for the readers of the journal in its current presentation. Please find enclosed our point-to-point reply to the criticisms. 

  • The manuscript title has been changed as follows: “Vitiligo-like lesions and COVID-19: case report and review of vaccination- and infection-associated vitiligo”.
  • We provided additional data on prevalence of vitiligo, based on regional distribution, accordingly, we modified the text of the manuscript in the related section and assessed the cases for country of origin by adding a dedicated column in table 1.
  • Terms and limits of our literature search has been included in the related paragraph (Introduction).
  • Similarly to the title, the paragraph has been changed according with your suggestions.

Reviewer 3 Report

The authors in this case study described vitiligo- like lesions after immunization with COVID-19 vaccination. Although, the study seems to be interesting and could potentially be of fundamental importance, the study design is not impressive. Authors need to address following major concerns.

1. The case history is too small.

2. Enough background information on the mechanism of vitiligo is not stated.

3. How can the authors justify the clinical relevance of the study?

4. Study design is too observational.

Author Response

Dear Editor

first of all, we would like to thank you for the valuable comments, aimed to improve the quality of our manuscript. We tried to fit the questions arisen during the review process and we hope that the paper could be considered of interest for the readers of the journal in its current presentation. 

  • The patient was a young girl with no history of comorbidities having an overall good general health. The onset of vitiligo represented the only relevant disease in its history and, because of the pandemic, also the number of visits at hospital were limited. Anyway, we improved the report of the case with additional notes.  
  • The discussion section has been revised and integrated with updated information on pathogenesis of vitiligo. We are confident that it could result in the improvement of the chapter.
  • In our opinion, the interest of our manuscript is represented by the report of an additional case, not so frequent, of vitiligo onset after COVID 19 vaccination and the review of the pertinent literature. We think that the information included should be useful for clinicians reading the journal.
  • Because of the relative rarity of the phenomenon, with few cases reported, and the still not fully explained pathogenesis of vitiligo, in our opinion this type of study better fit the attention and the needing of the physician focused on clinical practice. With this aim, we carefully revised the literature and some interesting points have emerged.

Round 2

Reviewer 3 Report

Can the authors discuss the clinical importance of the study done in the manuscript?

Author Response

Dear Reviewer, 

we thank you again for the further support in improving the quality of our manuscript. According with your suggestion, we added a sentence at the end of the Discussion section, explaining our idea on the clinical relevance of the whole manuscript.

Hoping that in its actual form the manuscript could be considered suitable for publication, we send you our best regards.